

# Labeled-free quantitative proteomic analysis of cervical squamous cell carcinoma identifies potential protein biomarkers

Hua Bai and Hongyun Zheng

Department of Clinical Laboratory, Renmin Hospital of Wuhan University, Wuhan, Hubei province, China

## ABSTRACT

**Background:** Cervical cancer remains a prevalent cancer among women, and reliance on surgical and radio-chemical therapies can irreversibly affect patients' life span and quality of life. Thus, early diagnosis and further exploration into the pathogenesis of cervical cancer are crucial. Mass spectrometry technology is widely applied in clinical practice and can be used to further investigate the protein alterations during the onset of cervical cancer.

**Methods:** Employing labeled-free quantitative proteomics technology and bioinformatics tools, we analyzed and compared the differential protein expression profiles between normal cervical squamous cell tissues and cervical squamous cell cancer tissues. GEPIA is an online website for analyzing the RNA sequencing expression data of tumor and normal tissue data from the TCGA and the GTEx databases. This approach aided in identifying qualitative and quantitative changes in key proteins related to the progression of cervical cancer.

**Results:** Compared to normal samples, a total of 562 differentially expressed proteins were identified in cervical cancer samples, including 340 up-regulated and 222 down-regulated proteins. Gene ontology functional annotation, and KEGG pathway, and enrichment analysis revealed that the differentially expressed proteins mainly participated in metabolic pathways, spliceosomes, regulation of the actin cytoskeleton, and focal adhesion signaling pathways. Specifically, desmoplakin (DSP), protein phosphatase 1, regulatory (inhibitor) subunit 13 like (PPP1R13L) and ANXA8 may be involved in cervical tumorigenesis by inhibiting apoptotic signal transmission. Moreover, we used GEPIA database to validate the expression of DSP, PPP1R13L and ANXA8 in human cancers and normal cervix.

**Conclusion:** In this study, we identified 562 differentially expressed proteins, and there were three proteins expressed higher in the cervical cancer tissues.

The functions and signaling pathways of these differentially expressed proteins lay a theoretical foundation for elucidating the molecular mechanisms of cervical cancer.

Corresponding author
Hongyun Zheng,
shenzheng2008@163.com

## INTRODUCTION

Cervical cancer stands as the fourth most frequently diagnosed cancer and the fourth leading cause of cancer-related deaths among women, with an estimated 604,000 new cases and 342,000 deaths globally in 2020 (*Sung et al., 2021*). Although incidence and mortality rates have declined in most regions worldwide over the past decades, substantial disparities in cervical cancer mortality still exist between impoverished and non-impoverished areas (*Siegel, Miller & Jemal, 2019*). Human papillomavirus (HPV) is a necessary but insufficient cause of cervical cancer (*Walboomers et al., 1999*), with other contributing factors including certain sexually transmitted infections, smoking, aging, and prolonged use of oral contraceptives (*Burd, 2003*). Despite significant advances in preventive vaccination, cytological screening, and cervical cancer treatment, a decline in survival rates due to late diagnosis and inadequate treatment remains a paramount challenge. Although HPV vaccination can play a positive preventive role (*Zou, Huang & Li, 2022*), there is still no optimal therapeutic drug for cervical cancer to date. Thus, a deeper understanding of the proteomic characteristics of cervical cancer is imperative, actively seeking target molecules in the occurrence, development, and diagnosis of cervical cancer, providing a basis for the study of cervical cancer mechanisms and personalized medicine.

Proteomics plays a crucial role in elucidating the pathogenic mechanisms of tumors (*Collinson et al., 2024*). This study employs labeled-free quantitative proteomics technology to identify differentially expressed proteins (DEPs) in cervical cancer groups compared to normal groups. Label-free protein quantitation is a mass spectrometry-based method (*Makepeace et al., 2024*) wherein protein peptides are enzymatically digested and analyzed, obviating the need for costly stable isotope labels as internal standards and only analyzing the mass spectral data produced during large-scale protein identification (*Zou, Huang & Li, 2022*). This process aids in systematically understanding the proteome after the onset of cervical cancer. Utilizing bioinformatics, this study analyzes the major functions and signaling pathways of these DEPs, exploring their molecular mechanisms in the onset of cervical cancer.

In this research, we use mass spectrometry-based quantitative proteomics methods to obtain the proteomic spectrum of cervical squamous cell carcinoma, achieving comprehensive analysis of differentially expressed proteins through GO/KEGG analysis and Gene Set Enrichment Analysis (GSEA). Our analysis reveals 562 up-regulated or down-regulated proteins compared to the control. The proteins DSP, PPP1R13L and ANXA8 are selected for further analysis. Immunohistochemistry and western blotting of cervical squamous cell carcinoma tissue slices with antibodies against DSP, PPP1R13L and ANXA8 is conducted, hypothesizing these proteins' fundamental roles in the regulation and progression of cervical cancer cells.

## MATERIALS AND METHODS

### 4D-label free quantitative proteomics technology

In recent years, the rapid development of 4D-label free quantitative proteomics technology has brought a new breakthrough for the study of biopharmaceutical mass spectrometry.

The technique quantifies protein expression levels by analyzing ionized peptides or protein fragments in a protein sample. It is based on mass spectrometry and allows for highly sensitive Quantitative analysis, which uses high-resolution mass spectrometry instruments and advanced data processing algorithms to detect and quantify trace amounts of proteins in samples, providing more accurate and reliable quantitative results. 4D-label free technology combines efficient sample preparation method with automated mass spectrometry platform, which can analyze a large number of samples quickly and improve the analysis efficiency and sample processing ability.

## Clinical samples

The specimens were collected from five pairs of cervical squamous cell carcinoma and corresponding paracervical normal tissues from Renmin Hospital of Wuhan University. All samples were stored at −80 °C and a preliminary histopathological diagnosis was made. Cancerous tissues and normal cervical tissues were used for proteomic analysis. This study was approved by the Ethics Committee of Wuhan University Renmin Hospital (WDRY2022-K173). We have obtained written informed consent from study participants.

## Protein extraction

An appropriate amount of SDT lysate was added into the tissue or microbial body precipitation, which was then transferred to Lysing Matrix A tube and homogenized by MP homogenizer (24 × 2, 6.0 M/S, 30 s, twice). After ultrasound, it was placed in a boiling water bath for 10 min. A total of 14,000 $g$ was placed in a centrifuge for 15 min, and the supernatant was filtered with a 0.22 µm centrifuge tube to collect filtrate. The protein was quantified by the BCA method. Samples were stored at −80 °C.

## SDS-PAGE electrophoresis

A total of 20 µg protein from each sample was added to 6X loading buffer, placed in a boiling water bath for 5 min, 12% SDS-PAGE electrophoresis (constant pressure 250 V, 40 min), and Coomasil bright blue staining.

## FASP enzymatic hydrolysis

A total of 100 ug protein solution was taken for each sample, and DTT was added to the final concentration of 100 mM, boil water for 5 min, and cool to room temperature. A total of 200 µL UA buffer was added and mixed well, then transfered to a 30 kD ultrafiltration centrifuge tube to be centrifuged 12,500 $g$ for 15 min, then the filtrate was discarded (this step was repeated once). Then, a total of 100 µL IAA buffer (100 mm IAA in UA) was added and shaken at 600 rpm for 1 min, react at room temperature without light for 30 min, centrifuge 12,500 $g$ for 15 min. A total of 100 µL UA buffer should be added and 12,500 g should be centrifuged for 15 min. This step was repeated twice. A total of 100 µL 50 mM NH4HCO3 solution was added 12,500 $g$ was centrifuged for 15 min, and this step was repeated twice. The collection tube was replaced, 40 µL Trypsin buffer (4 µg Trypsin in 40 µL 50 mM NH4HCO3 solution) was added, and oscillated at 600 rpm for 1 min, and placed at 37 °C for 16–18 h. A total of 12,500 $g$ was centrifuged for 15 min; then, 40 µL 50 mM NH4HCO3 solution was added, 12,500 $g$ from the center for 15 min, and the filtrate

was collected. The peptide was desorted using C18 Cartridge, lyophilized, redissolved in 40 μL 0.1% formic acid solution, and quantified (OD280).

## Mass spectrum analysis

### Easy nLC chromatography

Each sample was separated using the nanoliter flow rate Easy nLC system. Buffer solution A is 0.1% formic acid aqueous solution, and solution B is 0.1% formic acid acetonitrile aqueous solution (80% acetonitrile). The chromatographic column was balanced with 100% liquid A, and the samples were separated by automatic injector into analytical column (Acclaim PepMap RSLC 50 um × 15 cm, nano viper, P/N164943; Thermo Fisher Scientiific, Waltham, MA, USA). The flow rate was 300 nL/min.

### Mass spectrometry identification

The samples were separated by chromatography and analyzed by Orbitrap Exploris 480 mass spectrometer. The analysis time was 90 min (according to the specific experimental method), the detection method is positive ion, the scanning range of parent ion is 350–1,200 m/z, the primary mass spectrometry resolution is 120,000, the AGC target is 300%, and the primary Maximum IT is 50 ms. The mass charge ratio of peptides and peptide fragments was collected according to the following methods: Data Dependent Mode was set to Cycle Time, Cycle Time was set to 1.5 s, MS2 Activation Type was set to HCD. The Isolation window is 1.6 m/z, the AGC target is 75%, the secondary mass spectrometry resolution is 15,000, the Microscans are 1, the secondary Maximum IT is 35 ms, and the ion dynamic exclusion time is 30 s. Normalized Collision Energy is 33%.

### Mass spectrometry file processing

Maxquant is a leading qualitative and quantitative algorithm for proteomics and has gradually become one of the standard solutions in the field of proteomics in recent years. In proteomics research, high-resolution mass spectrometers can acquire raw data in two ways: data-dependent acquisition (DDA) and data-independent acquisition (DIA). The protein quantitative analysis based on label-free technology generally adopts the DDA method to collect the original data, which is the shotgun method in a common sense. In the DDA mode acquisition mode, the mass spectrometer first collects MS1 data, and then collects several MS2 data. In this experiment, the non-label quantitative method based on MS1 data integration is adopted. Since MS1 has a very high data density, the peptide feature of the original data needs to be intelligently identified before the data integration is calculated. The working process of Maxquant software is to identify the peptide feature first, and then calculate the strength value by graphically integrating.

Maxquant has undergone several upgrades and is now highly integrated and intelligent. In the data analysis stage of label-free experiment, the intelligent one-click operation is fully realized. After the mass spectrometry of the original document collection, only need to set the grouping, database and post-translation modification type, you can directly operate. Maxquant not only provides non-label quantitative strength data, but also automatically performs database matching to obtain qualitative results for proteins. In Maxquant's qualitative matching, the algorithm uses FDR principle to filter the data and

obtain highly reliable qualitative results. In addition, qualitative sequence information and quantitative information can be automatically correlated in the software to form a unified tabular file, which is convenient for subsequent analysis. Protein abundance was calculated based on normalized spectral protein intensity (LFQ intensity).

### GO function annotation and KEGG path analysis (*Ding et al., 2022*)
Gene ontology (GO) function annotation analysis was performed on all identified proteins, and the GO function of cell components, biological processes, and molecular functions corresponding to all proteins were analyzed. see http://www.geneontology.org for more information. Genomic (KEGG) pathway analysis was performed on the selected DEPs using the online analysis software Omicsbean (http://www.omicsbean.cn/). The data is normalized based on Z-Score and then clustered. The cluster analysis was drawn using R script (Complex-Heatmap 2.4.3).

### GSEA analysis (*Li et al., 2024*)
Gene set enrichment analysis (GSEA) was performed using GSEA v.4.2.3 (Broad Institute, Cambridge, MA, USA, https://www.gsea-msigdb.org/gsea/). We set parameters and run enrichment tests on a gene-set database to obtain permutation number (1,000) and permutation type (phenotype).

### Immunohistochemistry and western blotting
We analysed the protein expression of three molecules (DSP, PPP1R13L and ANXA8) by immunohistochemistry and western blotting in normal and cervical cancer tissue.

### Statistical analysis
Statistical analysis was performed with Statistical Program for Social Sciences (SPSS) (SPSS, version 22.0; SPSS Inc., Armonk, NY, USA). The significant difference was analyzed with a t-test between two groups, $P$-value < 0.05 was considered statistically significant. Proteins with $P$-value < 0.05 and fold changes (FC) >1.5 were considered as differentially expressed proteins.

### GEPIA analysis
GEPIA is an online website for analyzing the RNA sequencing expression data of tumor and normal tissue data from the TCGA and the GTEx databases (http://gepia.cancer-pku.cn/) (*Tang et al., 2017*). We used the GEPIA database to validate the expression of DSP, PPP1R13L and ANXA8 in human cancers and normal cervix.

## RESULTS

### Differential expression of the proteins
In the analysis of significant differences in the quantitative results, we first filter the data within the sample group where at least half of the repeated experimental data are non-null values for differential comparative analysis. Proteins meeting the criteria of an expression difference multiple greater than 1.5 and a $P$-value less than 0.05 are considered differentially expressed proteins (DEPs). Compared to normal results, there are a total of

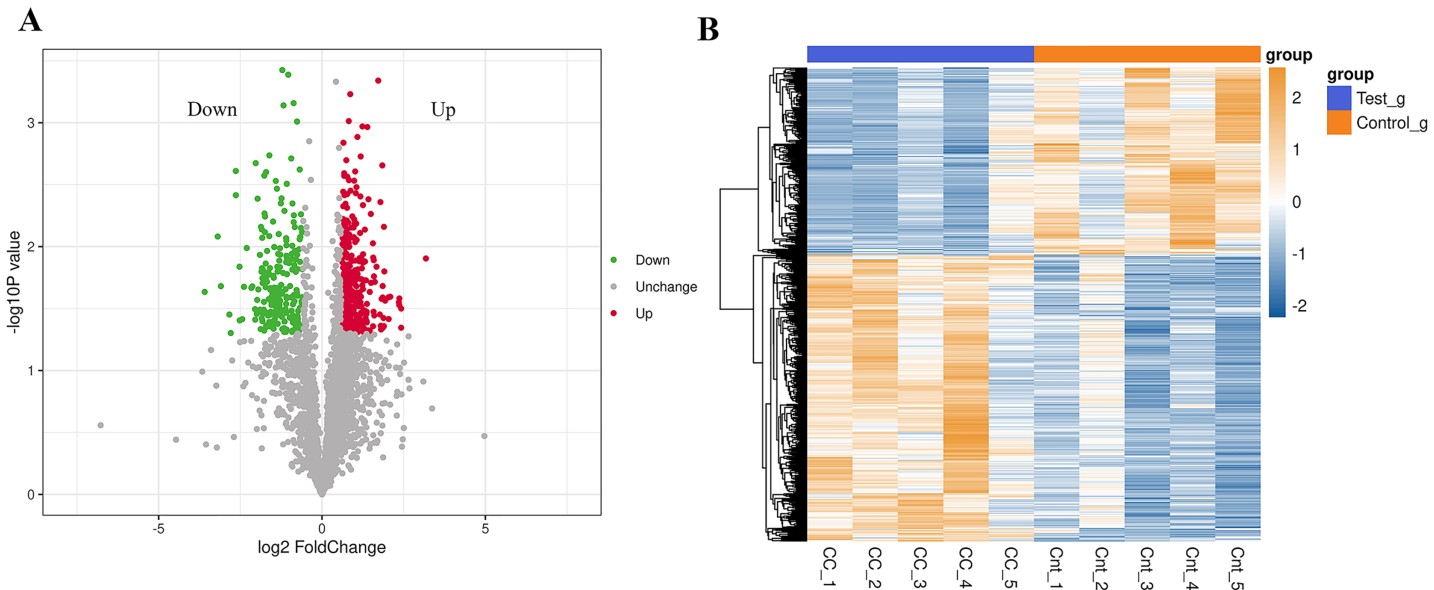

**Figure 1** **Differential expression of the proteins.** (A) A volcano plot is drawn using the protein expression difference multiple and *P*-value between the two groups of samples, showing the significant differences in data between the two groups of samples. Significantly up-regulated or down-regulated DEPs are represented in red or green, while proteins with no significant change are shown in black. (B) Compared with the control group, hierarchical cluster analysis of DEPs in the cervical cancer group shows that DEPs are up-regulated (orange) or down-regulated (blue). Ctr1~5 are five biological replicates of the normal group. CC1~5 are five biological replicates of the cervical cancer group. Each row represents the relative expression of a protein in 10 samples, and each column represents the relative expression of differentially abundant proteins in each experimental group. (*P* < 0.05, log2 fold change ≥1.0 or ≤−1.0). X-axis: log2 converted difference multiple values; Y-axis: −log10 transformed *P*-value.

562 DEPs in the cervical cancer group, with 340 up-regulated and 222 down-regulated (Fig. 1A). The top 20 up-regulated and down-regulated DEPs are shown in Table 1. Cluster analysis found significant differences between the cervical cancer group and the normal group, and the data patterns of five samples in each group have high similarity, proving that some protein levels indeed differ between the two groups (Fig. 1B).

## GO analysis of the differentially expressed proteins

To understand the function of different proteins and the signaling pathways they are involved in, we annotated 562 different proteins using GO and KEGG. GO analysis includes biological process (BP), cellular component (CC), and molecular function (MF) (Fig. 2A). The overall functional enrichment characteristics of all different proteins was revealed based on GO functional entries and displayed in the form of a bubble chart (Fig. 2B). To further understand the characteristics of these differentially expressed proteins, GO classification and enrichment analysis were conducted (Fig. 2C). In biological processes, the terms enriched the most are viral transcription, mRNA export from the nucleus, platelet degranulation, and extracellular matrix organization. In molecular function, proteins are mainly enriched in serine-type endopeptidase inhibitor activity, actin filament binding, structural constituents of the cytoskeleton, *etc*. In cellular components, most proteins are involved in focal adhesion, extracellular space, endoplasmic reticulum lumen, et al. To further understand the functions of the DEPs, we
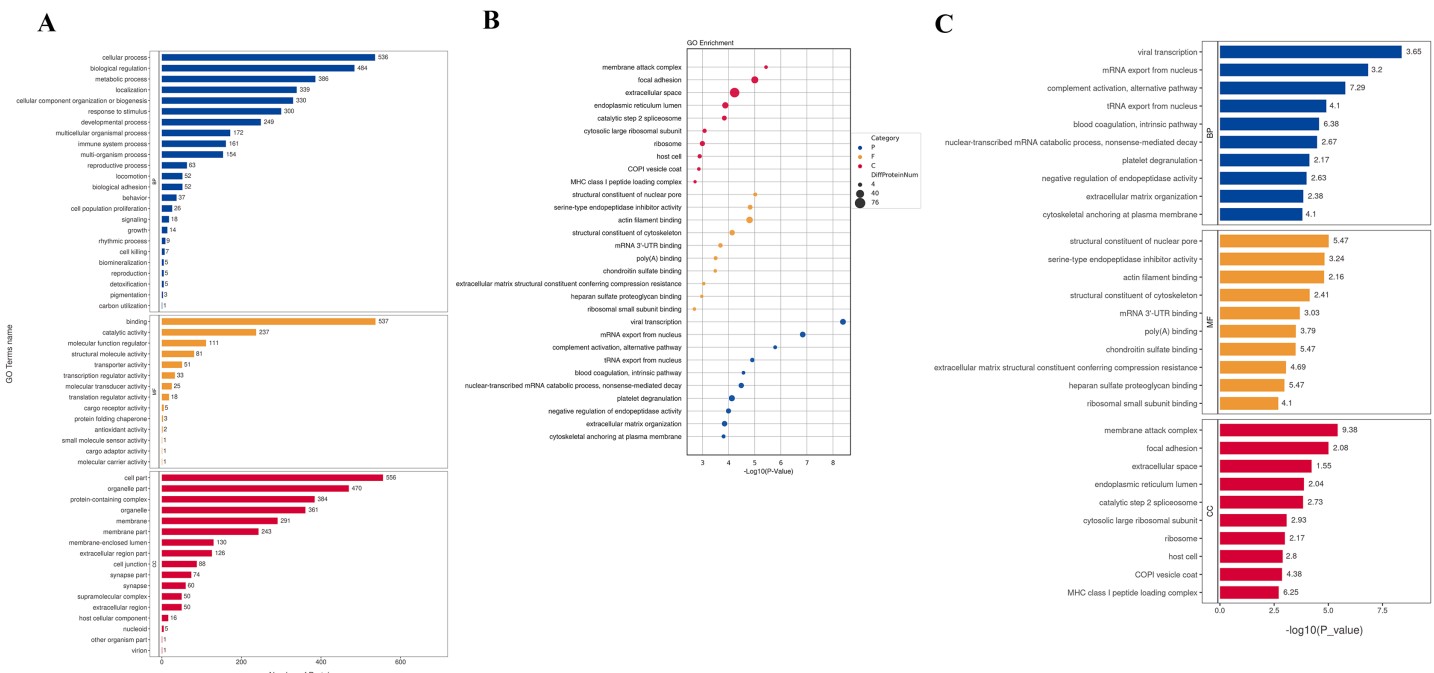

**Figure 2 GO analysis.** (A) GO channel enrichment statistical histogram (top 20) (B) GO enrichment analysis bubble diagram of DEPs. The largeness of the bubble indicates more DEPs. The color of the circle indicates different GO functional categories, and the size of the circle indicates the number of different proteins contained in the function. Ordinate :GO function name, ordinate :−log10 transformed *P* value. (C) The bar chart describes the distribution of GO major terms in the three GO categories. The horizontal coordinate represents the *P*-value after the −log10 transformation of the gene, and the vertical coordinate represents the GO function name of the gene, which is marked with Rich Factor value in the blue, red and yellow components respectively.               

analyse the top 10 most enriched items in biological processes, cellular components, and molecular functions in each category for up-regulated or down-regulated differential protein enrichment analysis. The functions of these differential proteins may provide clues and theoretical foundations for further elucidating the pathogenesis and development of cervical cancer (Fig. 3).

## KEGG analysis

The Kyoto Encyclopedia of Genes and Genomes (KEGG) is one of the databases commonly used for pathway research. It includes pathway information from various aspects such as metabolism, genetic information processing, environmental information processing, cellular processes, biological systems, human diseases, and drug development. Through KEGG pathway annotation for proteins with significant differential expression, it aids in understanding the metabolic or signal pathways these proteins may participate in. This reveals a series of changes from the cell surface to the cell nucleus, uncovering a series of biological events and acting factors involved in the process and indicating the potential biological consequences of the interruption or change in a particular process. Based on KEGG annotation results, the differential proteins in the pathways are sorted, displaying the TOP 20 KEGG pathway annotation results as shown in Fig. 4A. The KEGG pathway enrichment analysis method is similar to GO enrichment analysis. It uses the KEGG pathway as a unit, all qualitative proteins as a background, and analyzes and calculates the

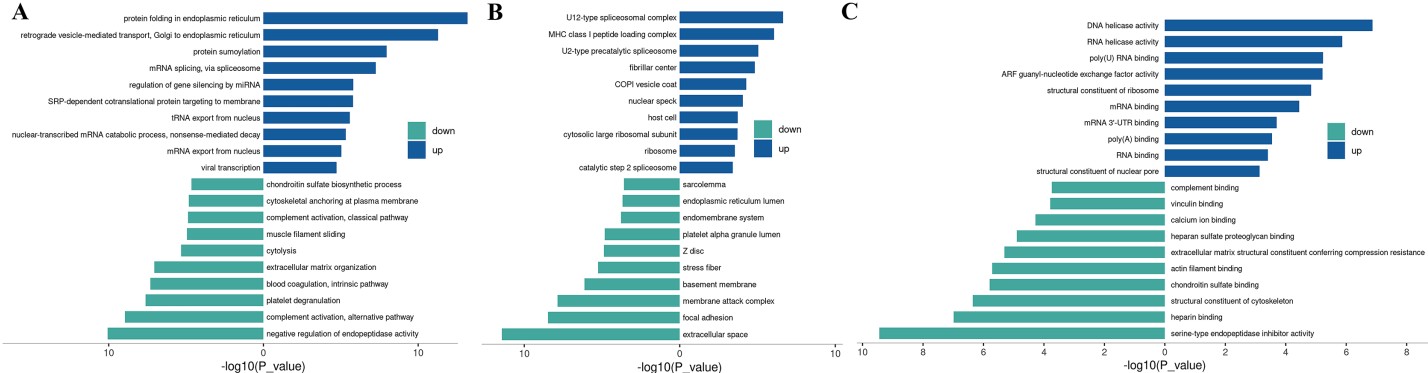

**Figure 3 Up-regulated and down-regulated differential protein GO enrichment analysis.** (A) Up-regulated protein GO function enrichment histogram (cc); (B) up-regulated protein GO function enrichment histogram (BP); (C) up-regulated protein GO function enrichment histogram (mf) Y-axis: significantly enriched GO analysis items; X-axis: *P*-value after −log10 transformation. The longer the column, the more significant the richness of the project. The blue and green column colors indicate the up-and-down changes in protein.

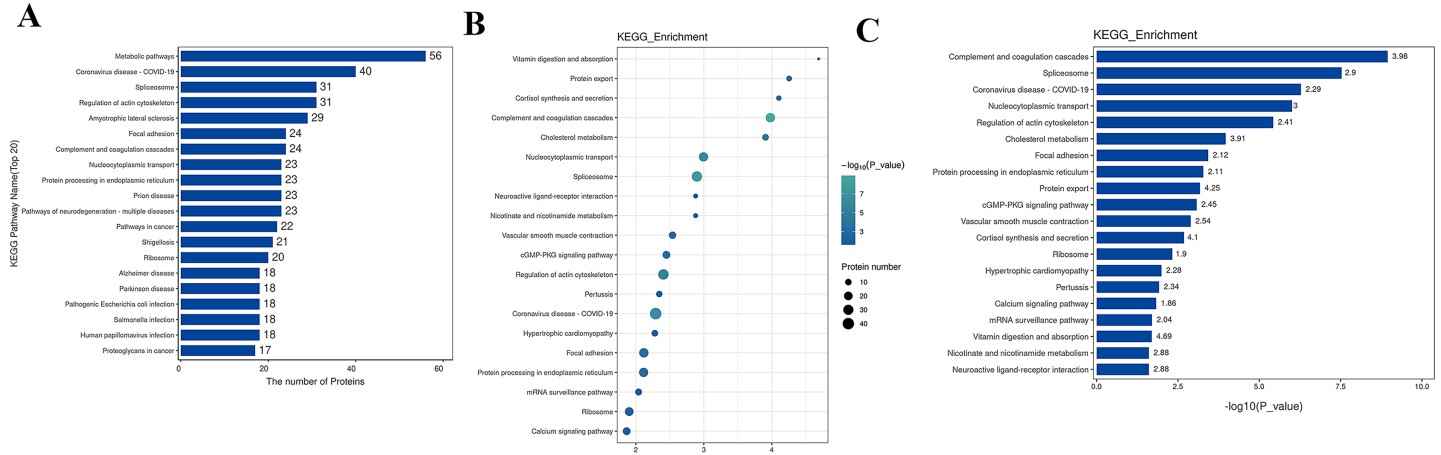

**Figure 4 KEGG analysis results.** (A) KEGG pathway enrichment histogram (top 20). Included are metabolic pathway, coronavirus disease, splicesome, actin cytoskeleton, amyotrophic lateral sclerosis, *etc.* Ordinate: KEGG pathway name; Ordinate: −log10 transformed *P*-value. The numbers in the column represent the Rich Factor values. (B) KEGG pathway concentration statistical bubble chart (top 20): including the vitamin digestion, protein export, cortisol synthesis, complement and coagulation cascades, nucleocytoplasmic transport, protein processing in endoplasmic reticulum, prion disease, pathways of neurodegeneration, multiple diseases, pathways in cancer, *etc.* The ordinate is the KEGG pathway name; abscissa: rich factor value; the color of the circle represents the −log10 transformed *p*-value; the darker the color, the smaller the *p*-value. (C) The KEGG Pathway Enrichment Bar Chart presents a series of related metabolic or signaling pathways, such as vitamin digestion and absorption, protein export cortisol synthesis and secretion, complement and coagulation cascades, cholesterol metabolism, nucleocytoplasmic transport, spliceosome, neuroactive ligand-receptor interaction, nicotinate and nicotinamide metabolism, vascular smooth muscle contraction, cGMP-PKG signaling pathway, and so on.

significant level of protein enrichment in each pathway through Fisher's Exact Test, thereby determining the significantly affected metabolism and signal transduction pathways. The results are shown in bubble charts (Fig. 4B) and bar charts (Fig. 4C). KEGG enrichment analysis results suggest that significantly expressed proteins play important roles in complement and coagulation cascades, spliceosomes, coronavirus disease, nucleocytoplasmic transport, regulation of the actin cytoskeleton, cholesterol metabolism,

*etc.* The functions of DEPs involved may provide clues and a theoretical basis for further clarifying the pathogenesis and development of cervical cancer.

## GSEA analysis

The GSEA (Gene Set Enrichment Analysis) is not limited to differentially expressed proteins and directly uses the expression levels of all genes for analysis, effectively compensating for the deficiencies of traditional enrichment analysis. GSEA can also be used to determine whether a certain pathway is activated or inhibited in a set of samples. GSEA operates by sorting all genes based on the degree of differential expression between two sample groups and then calculating the enrichment degree and significance at the top or bottom of the pre-defined gene set in this sorted list. The Enrichment Score (ES) reflects the degree of gene set members enriched at both ends of the gene-sorted list. The Normalized Enrichment Score (NES) is obtained by normalizing according to the size of the gene set, and the false discovery rate (FDR) value is calculated to control the false-positive rate. GSEA analysis for the GO term is performed, and the ES plot of the GO function with the highest NES value in the experimental group is shown in Fig. 5-1. The bubble chart is used to visualize the GSEA-GO enrichment results, selecting the top 20 functions by NES value and calculating the Gene_ratio of the GO function, sorted from largest to smallest. The most relevant are mRNA end processing regulation and RNA catabolic regulation, as shown in Fig. 5A. The ES plot of the GO function with the highest NES value in the experimental group, namely GOCC CAJAL BODY, is shown in Fig. 5C. For KEGG Pathway GSEA analysis, a bubble chart is used to visualize the GSEA-KEGG enrichment results, selecting the top 20 pathways by NES value and calculating the Gene_ratio of the KEGG pathway, sorted from largest to smallest, as shown in Fig. 5B. The most relevant are the ribosome and spliceosome. The ES plot of the KEGG pathway with the highest NES value in the experimental group, namely KEGG RIBOSOME, is shown in Fig. 5D.

## Structural domain, transcription factor analysis, and sub-cellular localization analysis

A protein domain represents a specific structure and independent functional region within a protein, serving as a fundamental unit for protein structure, function, and evolution. Compared to the vast number of proteins, the number of structural domains is limited, each encompassing a finite number of members. Exploring these domains is crucial for understanding the biological function and evolution of proteins. Typically ranging between 25 and 500 amino acids in length, structural domains are analyzed in this study using the InterPro database, a comprehensive resource that integrates protein families, domains, and functional sites. The functional structural domains of differential proteins are analyzed, and the annotated results are enriched. The top 20 enriched domains are visually represented through bubbles (Fig. 6A) and bar graphs (Fig. 6B). The domains with the highest Rich factor are: Guanine nucleotide exchange factor, anaphylatoxin, complement system domain, complement_C3_C4_C5, anaphylatoxin, complement system, serpin, conserved site RNA recognition motif domain, serpin family, serpin
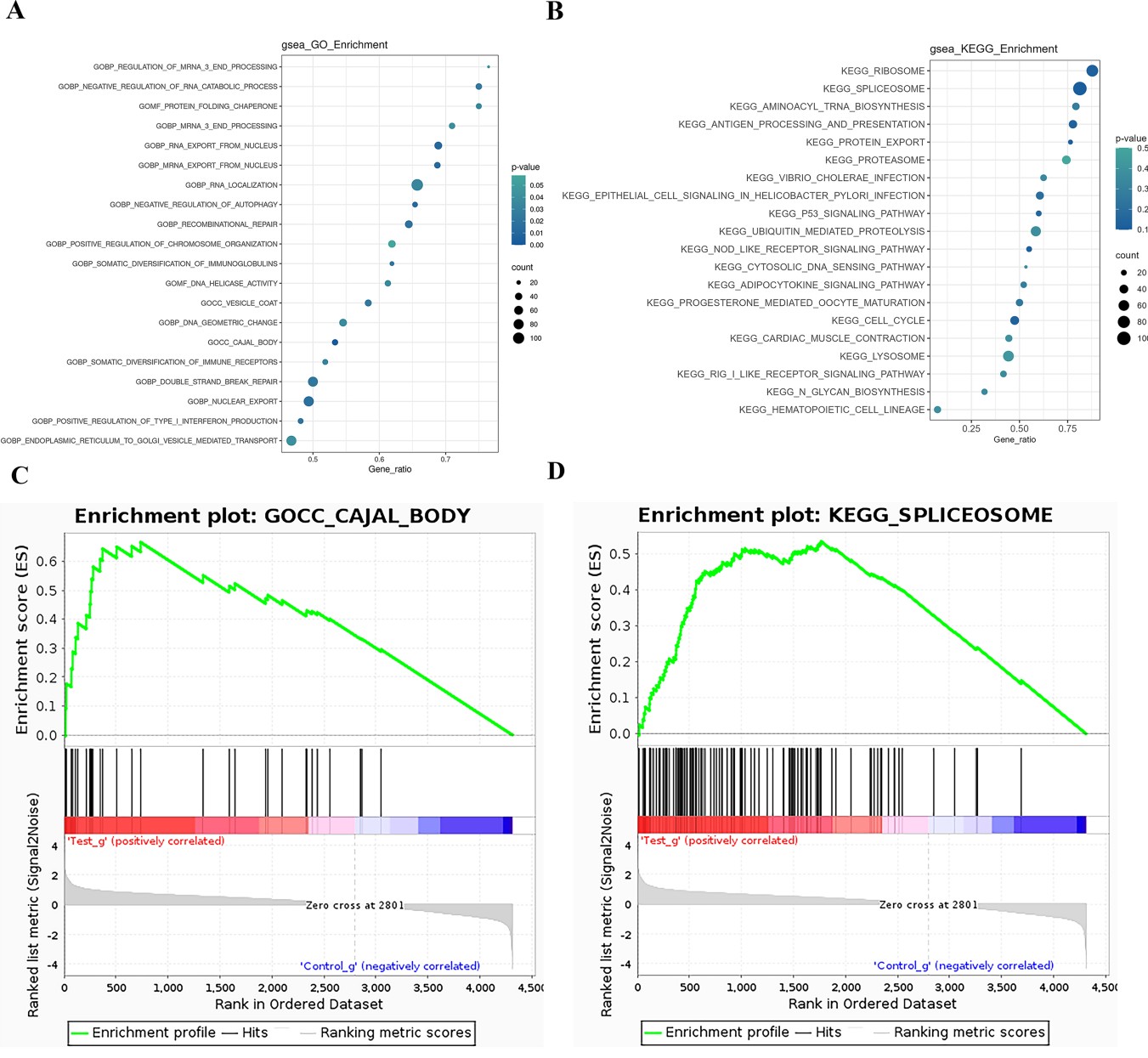

**Figure 5 GSEA analysis results.** (A) GSEA-GO enrichment analysis NES value top 20 bubble chart: Ordinate: GO function name, Abscissa: Gene_ratio, Circle size: the number of enriched genes in the GO function item; circle color: Enrichment significance: *P* value. (B) GSEA-KEGG enrichment analysis NES value top 20 bubble chart; Ordinate: KEGG pathway name, abscissa: Gene_ratio; circle size: the number of genes enriched in KEGG pathway entries; circle color: enrichment significance: *P* value. (C) GSEA-GO enrichment score curve graph; the GSEA-GO ES enrichment score curve graph is composed of three parts. The upper part is the enrichment score (ES) line graph, displaying the scores (running ES) throughout the scoring process. Peaks in the line graph represent the ES values of the protein function set; a larger absolute ES value indicates a higher degree of enrichment. A positive ES value signifies that the functional protein set is enriched in up-regulated proteins, while a negative ES value signifies enrichment in down-regulated proteins. Functional sets with peaks at the top or bottom are typically of greater concern. The middle section displays the position of members of the function set in the expression set list, with black vertical lines representing members of the function set. The colored bands, composed of members of the expression set arranged in a certain order, are red for up-regulated and blue for down-regulated. The lower part displays the rank-value distribution of all proteins. (D) GSEA-KEGG enrichment score curve graph: Similar to the GSEA-GO graph, the GSEA-KEGG ES enrichment score curve graph is divided into three parts. The upper part is the Enrichment Score (ES) line graph, displaying the ES values (running ES) throughout the scoring process. The middle section and lower parts have the same functionality as mentioned in GSEA-GO.

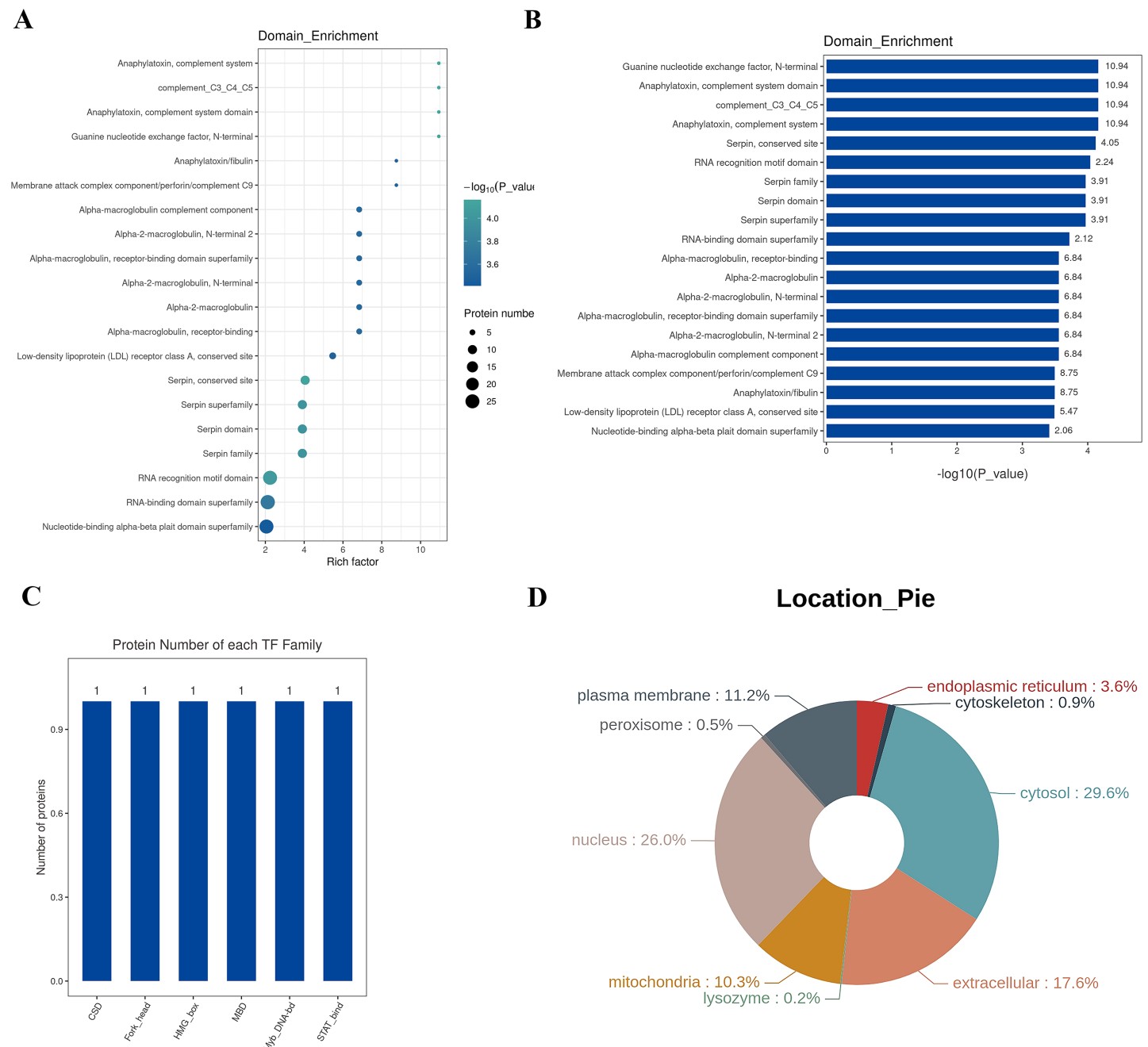

**Figure 6** **Structural domain, transcription factor analysis, and sub-cellular localization analysis.** (A) Enrichment classification statistical bubble chart for protein structure domain (top 20). The bubble chart shows that many proteins structure domains were involved in the study, such as guanine nucleotide exchange factor N-terminal, anaphylatoxin-complement system domain, complement_C3_C4_C5, anaphylatoxin-complement system, conserved site RNA recognition motif domain, serpin family, serpin domain, serpin superfamily, and the RNA-binding domain superfamily. Y-axis: structure domain name; X-axis: Rich Factor Value. The circle color represents the −log10 transformed *P*-value; a deeper color indicates a smaller *P*-value. The circle size indicates the number of differential proteins contained in this pathway. (B) Enrichment classification statistical bar chart for protein structure domain (Top 20). Y-axis: structure domain name; X-axis: −log10 transformed *P*-value. The number on the bar represents the Rich Factor value. (C) Top 10 TF family statistical chart; (D) protein subcellular localization distribution chart: Different colors represent different cellular locations, and the percentage indicates the proportion of proteins located there among all differential proteins.

domain, serpin superfamily, RNA-binding domain superfamily, *etc*. Transcription Factors (TFs) are proteins capable of binding specifically to certain sequences upstream of the 5′ end of a gene, ensuring the gene's expression at particular intensities, times, and spaces. Given the significance of TFs, in-depth analysis and annotation of these proteins are conducted. The Animal TFDB (Animal Transcription Factor Database) encompasses information about animal TFs and TF families, enabling the prediction of whether a protein under observation is a TF and to which TF family it belongs. The differential proteins are annotated as TFs, and the top 10 TF families are statistically analyzed, including Myb_DNA-bd, CSD, Fork_head, STAT_bind, MBD, and HMG_box families, as depicted in Fig. 6C.

Sub-cellular localization refers to the specific location within a cell where a protein or expressed product is found. The main Sub-cellular locations in eukaryotic cells include extracellular space, cytoplasm, cell nucleus, cell membrane, mitochondria, Golgi apparatus, endoplasmic reticulum, peroxisomes, cytoskeleton, nucleoplasm, nuclear matrix, and ribosomes, among others. Understanding the sub-cellular localization of proteins is pivotal for functional protein research. Proteins must be transported to the correct cellular organelle to participate in various life activities and effectively execute their functions. In this study, the software WoLF PSORT is employed for Sub-cellular localization analysis of differential proteins. The results are illustrated in Fig. 6D, where the top three Sub-cellular locations are cytosol (29.6%), nucleus (26.0%), and extracellular (17.6%).

## Protein-protein interaction network analysis (PPI)

Proteins do not exist independently within biological organisms; their functional roles are exerted through the regulation and mediation of other proteins. This regulatory or mediating action first requires binding or interaction between proteins. Research on the interactions between proteins and the networks formed by these interactions is essential for revealing protein function. For instance, highly aggregated proteins may possess the same or similar functions, and proteins with high connectivity may be key points affecting the entire system's metabolism or signal transduction pathways. In this project, we selected proteins with a *P*-value less than 0.05 and the most significant expression (50 up-regulated and 50 down-regulated) as target proteins for direct interaction network analysis (Fig. 7A). Furthermore, these 100 target proteins were used with all identified proteins in this project for indirect interaction network analysis (Fig. 7B).

## Verification of results of three differential proteins

Upon analysis of this proteome, among the top 20 significantly up-regulated and top 20 significantly down-regulated differential proteins (as shown in Table 1), this study identified three potential up-regulated proteins that may be related to the occurrence and development of cervical cancer: desmoplakin(DSP), protein phosphatase 1, regulatory (inhibitor) subunit 13 like (PPP1R13L), and ANXA8. In the immunohistochemical staining of DSP, PPP1R13L and ANXA8 in paracancer and cervical cancer tissues, the staining intensity of three proteins in cancer tissues was found to be enhanced, primarily

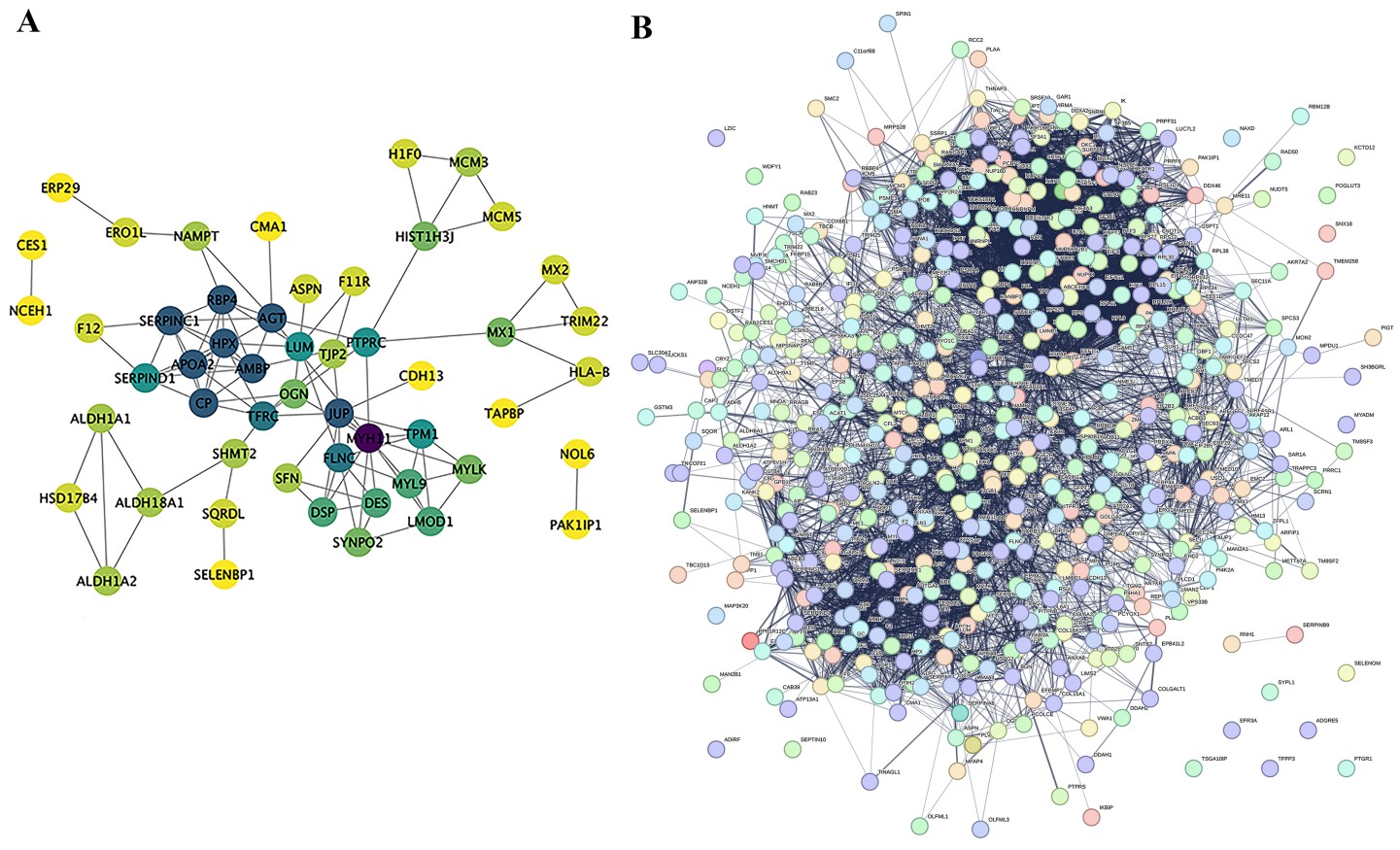

**Figure 7 Protein-Protein interaction network analysis (PPI).** (A) Direct interaction network of differential proteins; (B) interaction network of differential proteins with all other identified proteins. In the network: blue nodes represent the most dramatically up-regulated proteins. Cyan nodes represent the most dramatically down-regulated proteins. Yellow nodes represent differential proteins obtained through statistical testing using ANOVA P. Grey nodes represent other identified proteins. The node size indicates the degree value.

expressed in the cytoplasm and cell membrane compared with paracancer control (Fig. 8A). In order to further clarify the expression levels of these three proteins in cervical cancer tissue, then we tested the protein expression levels in cervical cancer and paracancer tissues. The results showed that the protein expression of all the three protein levels (DSP, PPP1R13L and ANXA8) increased in cervical cancer tissue compared with the paracancer control (Fig. 8B), and further statistical analysis showed that the expression levels of the three different proteins in cervical cancer were significantly increased (Fig. 8C, $P < 0.01$). To further verify the validity of the data, we used the GEPIA database to validate the expression of DSP, PPP1R13L and ANXA8 in human cancers and normal cervix. We found that the transcription levels of DSP, PPP1R13L and ANXA8 increased significantly (Fig. 8D). Therefore, the results suggest that these three proteins may play a significant role in the occurrence and development of cervical cancer.

## DISCUSSION

Cervical cancer remains a prevalent malignancy among women (*Hernández-Silva et al., 2024*), and reliance on existing therapeutic strategies continues to irreversibly impact

**Table 1  Top 20 upregulated proteins in cervical cancer group.**

| Protein IDs | Gene name | FC | p-value | Regulation |
|---|---|---|---|---|
| P15924 | DSP | 9.060566576 | 0.012458719 | UP |
| P25205 | MCM3 | 5.361321064 | 0.031581552 | UP |
| P14923 | JUP | 5.359627937 | 0.045097107 | UP |
| P13928 | ANXA8 | 5.157378248 | 0.026179129 | UP |
| P20591 | MX1 | 5.143888334 | 0.029632331 | UP |
| Q460N5 | PARP14 | 5.103731041 | 0.028108233 | UP |
| Q9C0C2 | TNKS1BP1 | 4.258501333 | 0.025380129 | UP |
| O15533 | TAPBP | 4.112167597 | 0.038363024 | UP |
| O15460 | P4HA2 | 4.100159913 | 0.025621014 | UP |
| P19971 | TYMP | 3.816846649 | 0.027280835 | UP |
| P28288 | ABCD3 | 3.748161909 | 0.015842479 | UP |
| P17844 | DDX5 | 3.718402644 | 0.006919522 | UP |
| Q9Y624 | F11R | 3.716451204 | 0.043790714 | UP |
| P02786 | TFRC | 3.680377824 | 0.037659713 | UP |
| Q14573 | ITPR3 | 3.642456382 | 0.045058574 | UP |
| P01889 | HLA-B | 3.620574734 | 0.025707783 | UP |
| P68366 | TUBA4A | 3.619224981 | 0.020790794 | UP |
| Q16706 | MAN2A1 | 3.592652315 | 0.002210281 | UP |
| P48729 | CSNK1A1 | 3.543471001 | 0.026387934 | UP |
| P34897 | SHMT2 | 3.499011034 | 0.046582261 | UP |

patients' life span and quality of life. Consequently, early diagnosis and further exploration into the pathogenic mechanisms of cervical cancer are paramount. Proteins, being executors of biological activities, offer a more intuitive insight into the proteomic expression spectrum of cervical cancer cells through the measurement and analysis of protein levels compared to genomic studies. Currently, mass spectrometry technology finds extensive application in clinical practice (*Keevil, 2013*), enabling further investigation into protein alterations throughout the cervical cancer pathogenic process using proteomic methods.

The mass spectrometry analytical method used in this experiment is the label-free quantification (label free) technique. This approach allows the analysis of an almost unlimited number of samples without introducing any labels, thereby maintaining low cost and minimizing sample preparation steps. This method is highly desirable for biomarker research. However, its poor reproducibility may necessitate analysis of numerous technical replicates, possibly leading to low accuracy in quantitative measurements. Despite this, compared to labeled strategies, the label-free method has been proven to possess the greatest dynamic range and highest proteome coverage (*Rozanova et al., 2021*).

In our research, we use the label-free quantification (label free) technique to obtain the proteomic spectrum of cervical squamous cell carcinoma, achieving comprehensive analysis of differentially expressed proteins through GO/KEGG analysis and gene set

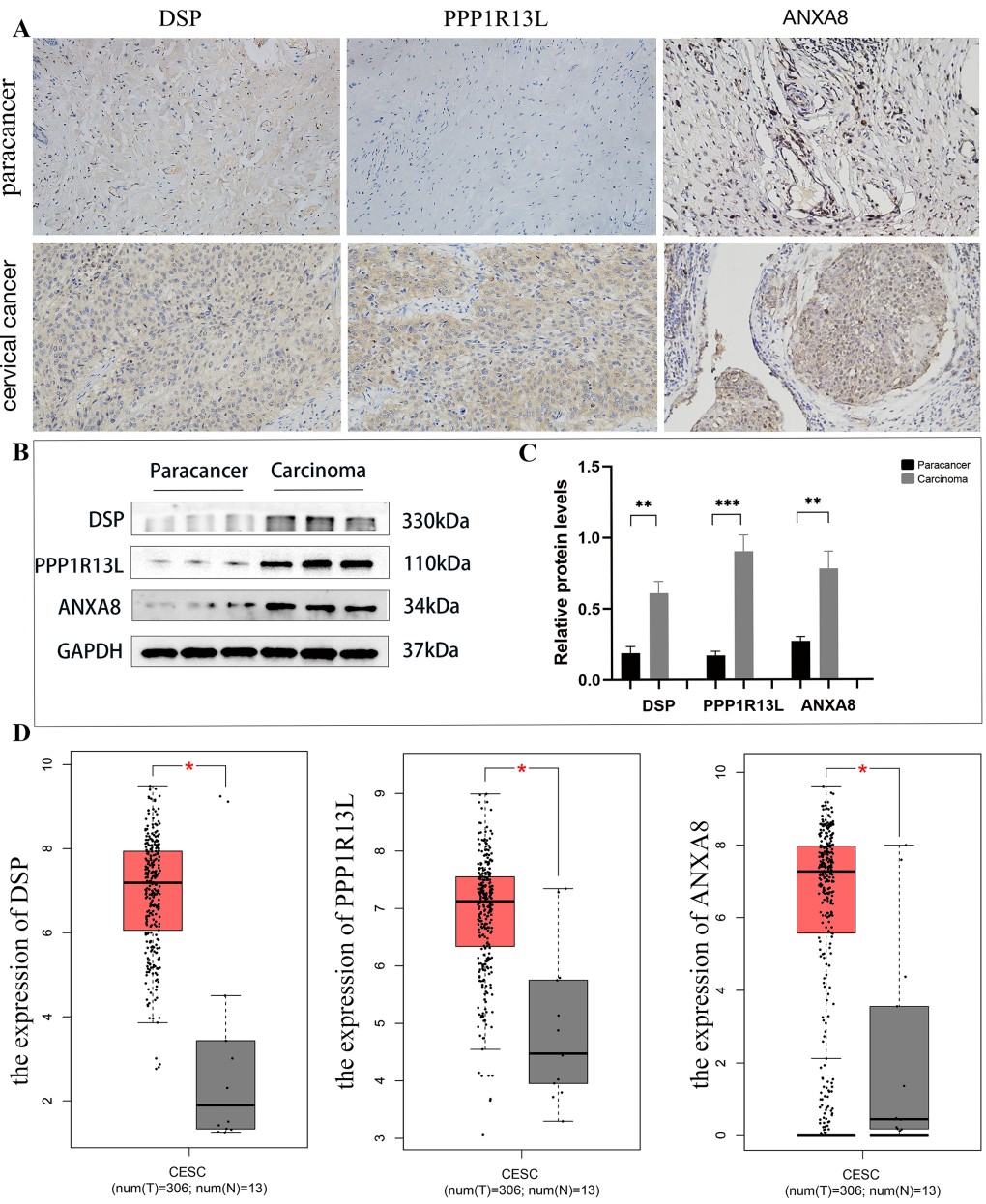

**Figure 8 Employing immunohistochemical and western blotting methods to detect the expression of DSP, PPP1R13L and ANXA8 in cervical cancer tissues and paired adjacent cancer tissues.** (A) Compared with the para cancer, the immunohistochemical results for DSP, PPP1R13L and ANXA8 demonstrated that their protein expression increased, (B) as well as overexpression of them in the cases of cervical cancer tissues by western blotting, (C) and there is a significant difference by the statistical results. $P = 0.0015$ & adjacent cancer tissues. (D) The GEPIA database was used to validate the expression of DSP, PPP1R13L and ANXA8 in human cancers and normal cervix, and it was found that the transcription levels of DSP, PPP1R13L and ANXA8 increased significantly ($P = 0.05$ & adjacent cancer tissues).

enrichment analysis (GSEA). Our analysis reveals 562 up-regulated or down-regulated proteins compared to the control. Based on the results of proteomics and the latest reported literature, we selected three different proteins from the 20 up-expressed protein

molecules for further analysis. The proteins DSP, PPP1R13L and ANXA8 are selected for further analysis. Immunohistochemistry and western blotting of cervical squamous cell carcinoma tissue slices with antibodies against DSP, PPP1R13L and ANXA8 is conducted, hypothesizing these proteins' fundamental roles in the regulation and progression of cervical cancer cells.

Desmoplakin (DSP) is involved in the organization of the desmosomal cadherin-plakoglobin complexes into discrete plasma membrane domains and in the anchoring of intermediate filaments to the desmosomes (*Andrei et al., 2024*). DSP serves as an obligate component of desmosomes that function as intercellular adhesion junctions and sites for intermediate filament attachment (*Boyden et al., 2016*). Besides maintaining the integrity of desmosomes, DSP also plays an important role as tumor suppressor (*Sano et al., 2021*) by regulating various signaling pathways in cancer cells. DSP has a tumor suppressive function through inhibition of Wnt/-catenin signalling pathway in human lung cancer (*Wang et al., 2019*). PPP1R13L gene, also known as iASPP (inhibitory member of the ASPP family, iASPP), is located in 19q13 (*Nexø et al., 2008*). As the third member of the TP53 apoptosis stimulating protein family, its role is to inhibit the anti-tumor effect of the other two members (ASPP1 and ASPP2), leading to abnormal cell proliferation and carcinogenesis (*Li et al., 2024*). Accordingly, it was generally believed that PPP1R13L as a novel oncogene highly expressed in a variety of tumor cells (*Yagudin et al., 2021*). Protein Phosphatase 1 Regulatory Subunit 13 Like (PPP1R13L) is a newly identified oncoprotein previously reported to inhibit the transcriptional activity of SP1 *via* a direct protein-protein interaction) (*Zhang et al., 2022*). The other study showed that High iASPP expression in the tumor cell cytoplasm is associated with poor outcomes of OSCC patients in terms of recurrence and survival, suggesting a role for iASPP as a novel biomarker and therapeutic target for oral cavity squamous cell carcinomas (OSCC) (*Kim et al., 2015*).

ANXA8 is a member of the annexin family and belongs to the superfamily of $Ca^{2+}$/phospholipid-binding proteins (*Zhou et al., 2021*). ANXA8 plays a crucial role in the principal communication pathways between $Ca^{2+}$ signal transduction and $Ca^{2+}$-regulated cell membrane dynamics in all eukaryotic organisms (*Lee et al., 2009*). These biological entities are closely associated with various diseases, such as cancer, diabetes, and autoimmune disorders. ANXA8 has been demonstrated to be intimately connected with various types of malignant cancers (*Zhang & Han, 2021*). According to proteomic Gene Ontology (GO) analysis predictions, ANXA8's primary functions in cervical cancer might include signal transduction, positive regulation of apoptosis processes, negative regulation of coagulation, protein oligomerization, and negative regulation of calcium ion chelation, among others (*Zhou et al., 2021*). Cancer-related articles have corroborated ANXA8's involvement in signal transduction functions. ANXA8 regulates the proliferation of human non-small cell lung cancer cells A549 through the EGFR-AKT-mTOR signaling pathway (*Yuan et al., 2021*). Over-expression of miR-140-3p, targeting ANXA8, inhibits bladder cancer cell proliferation, migration, invasion, and EMT (*Lee et al., 2009*). Downregulation of ANXA8 in the EGF-FOXO4 signaling pathway is involved in cholangiocarcinoma cell migration and tumor metastasis (*Hata, Tatemichi & Nakadate, 2014*). Regarding cervical cancer, reports demonstrate that ANXA8 is significantly

expressed in squamous cell carcinoma but not evidently in adenocarcinoma, suggesting its potential as a molecular marker to differentiate between cervical squamous cell carcinoma and adenocarcinoma (*Chao et al., 2006*). This proteomic data, derived from cervical squamous cell carcinoma cells, indeed detected the upregulation of ANXA8, hypothesizing that ANXA8, as an up-regulated protein, plays a significant role in cervical squamous cell carcinoma.

In summary, the above three molecules DSP, PPP1R13L and ANXA8 may play important roles in related tumors, but their role in cervical cancer is still unclear. Our group research found that DSP as well as PPP1R13L and ANXA8' expression were up-regulated by IHC method. Further study showed that the western blotting results showed that they were also overexpressed in cervical cancer tissues compared with the paracancer tissues, and further statistical analysis shows significant differences between the two groups. Thus results indicated that the three proteins DSP, PPP1R13L and ANXA8, studied above may play important roles in the occurrence or metastasis of cervical cancer, which can provide important theoretical supplements for the occurrence, development, and clinical transformation of cervical cancer. To further verify the validity of the data, we used GEPIA database to validate the expression of DSP, PPP1R13L and ANXA8 in human cancers and normal cervix. We found that the transcription levels of DSP, PPP1R13L and ANXA8 increased significantly.

This study employs label-free technology to explore the proteomics of cervical cancer tissue. In comparison with normal cervical tissue, differential proteins were identified, and further research was conducted on the molecular functions possessed by these differential proteins, their roles in biological processes, and their participation in molecular pathways. Additionally, three differential proteins, DSP, PPP1R13L and ANXA8, were selected, and their possible promoting roles in the occurrence and development of cervical cancer were speculated upon.

### Funding
The authors received no funding for this work.

### Competing Interests
The authors declare that they have no competing interests.

### Author Contributions
- Hua Bai conceived and designed the experiments, performed the experiments, analyzed the data, prepared figures and/or tables, authored or reviewed drafts of the article, and approved the final draft.
- Hongyun Zheng conceived and designed the experiments, performed the experiments, analyzed the data, prepared figures and/or tables, authored or reviewed drafts of the article, and approved the final draft.

## Human Ethics

The following information was supplied relating to ethical approvals (*i.e.*, approving body and any reference numbers):

This study was approved by the Ethics Committee of Wuhan University Renmin Hospital (WDRY2022-K173).

## Data Availability

The raw measurements are available in the Supplemental File.

## Supplemental Information

Supplemental information for this article can be found online at http://dx.doi.org/10.7717/peerj.17444#supplemental-information.

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
