# Peer review of "Labeled-free quantitative proteomic analysis of cervical squamous cell carcinoma identifies potential protein biomarkers"

_PeerJ, doi:10.7717/peerj.17444_

## Round 0.1 · original submission · Major Revisions

Please revise the manuscript as the reviewers suggested.

Reviewer 1 ·

Basic reporting

Clarity and Language: The manuscript is generally clear and uses professional English. However, certain sections require minor language polishing for enhanced clarity.
Structure and Context: The introduction and background provide a comprehensive context for the study. Literature is well-referenced and relevant, establishing the need for this research.
Figures and Data Presentation: Figures and tables are appropriately used to present data. However, some figures could benefit from more detailed legends for better understanding.

Experimental design

Originality and Scope: The research presents original work within the scope of the journal, focusing on identifying potential protein biomarkers in cervical squamous cell carcinoma using labeled-free quantitative proteomics.
Research Question and Knowledge Gap: The research question is clearly defined and addresses an identified knowledge gap in the early diagnosis of cervical cancer.
Methodology: The methods are rigorously described with sufficient detail for replication. However, some technical aspects of the mass spectrometry analysis could be elaborated for clarity.

Validity of the findings

Data Analysis: Statistical methods are sound, and data analysis appears robust. However, a more comprehensive discussion on the limitations of label-free quantification, particularly its reproducibility and accuracy, is warranted.
Interpretation of Results: The conclusions are well-linked to the research question, but they somewhat overstate the implications of the findings. The potential roles of identified biomarkers in cervical cancer pathogenesis are discussed, but the direct clinical relevance of these biomarkers needs more exploration.
Relevance and Impact: The study contributes meaningfully to the understanding of cervical cancer proteomics, but its direct applicability in clinical settings requires further investigatio

Additional comments

Ethical Considerations: Ethical approvals and sample collection procedures are appropriately documented.
Reproducibility and Data Sharing: The manuscript adheres to data sharing policies, but more detailed information on raw data access could enhance reproducibility.
Future Work: Suggestions for future studies include exploring the clinical applicability of these biomarkers and validating their roles in cervical cancer progression in larger, diverse cohorts.

While the study presents significant findings in the field of cervical cancer proteomics, the direct clinical applicability of these biomarkers needs further validation. Additionally, minor language polishing and a more critical discussion of the methodological limitations would strengthen the manuscript.

·

Basic reporting

The submitted study has a clear aim, the MS-proteomics analysis of cervical cancer differential protein expression in comparison with normal tissue. The work is correctly embedded into the background of the field, the clinical requirements for improved therapy in contrast to HPV vaccination. In this background a precise proteomics study is highly valuable containing possible bioinformatic analysis on protein-protein interactions and relevant pathways. The work is written in clear, unambiguous, professional English language. The literatures are well referenced and are relevant. The manuscript has a usual scholar structure, the figures are relevant of acceptable quality. One comment might be mentioned in this regard: The figures showing charts, pathways expression and enrichment comparisons require magnification on the readers´ computer or display to see important information details. The figures must be of enough resolution to give this information in increased magnification. In accordance with the requirements of the journal, raw data are supplied.

Experimental design

The submitted work is an original primary research report, as mentioned above, the research question is meaningful and the achieved results provide significant contribution to the knowledge in the field. The investigation is rigorously designed and performed, the methods are described with sufficient detail and
information to replicate. The required ethic approval is present.

Validity of the findings

The submitted work is direct analysis of patients tissue material protein investigation, where a limited access to source material and costs of the used methods must be taken into account. In this relation the use of the number of used samples are appropriate to evaluate the outcomes. All underlying data have been provided, they are robust, statistically sound and controlled. The conclusions are well stated, linked to original research question and limited to supporting results.

Additional comments

Figure 2. Please use “A”, “B” and “C” in the Figure caption instead of “1”, “2” and “3”.

·

Basic reporting

This paper is well prepared and its language is clear and the topic of this project is important in the medical field. The biomarkers of cervical cancer are still controversial matter and many other studies tried to find out the proper one and this piece of work light shaded on this issue.

The hypothesis would be more solid if the work include HPV genotyping as there are few recent studies found out a relationship between biomarkers and HPV.

Experimental design

I wish the statistics would be more stringent in p value and in fold changes (ex: 0.01 for p value and 2 for fold change). This will lead to decrease the number of up-and down regulated biomarkers and focus on the real significant one.
I would recommend to use the Cytoscape software to cluster these markers into groups by using MCODE app then you can analyze the pathogenicity for each cluster and if it is related cervical caner.
In immunohistochemistry test for marker ANXA8 statistics is needed and determine the specificity and sensitivity.

Validity of the findings

I wish the authors cite other proteomics biomarkers work (labelled and non-labelled) in discussion section to validate their work and to see if they agree with them in specific biomarkers to narrow the scope on them.

No need for the markers SHMT2 and JUP figures as they derived from human protein atlas database and the line in introduction "Immunohistochemistry of cervical squamous cell carcinoma tissue slices with antibodies against SHMT2, JUP, and ANXA8 is conducted, hypothesizing these proteins’ fundamental roles in the regulation and progression of cervical cancer cells" should be revised as only ANXA8 was utilized in IHC work.

---

## Round 0.2 · Minor Revisions

The Section Editors of PeerJ have commented and said:

We highlight a number of issues that should be addressed prior to acceptance.

The authors have performed a proteomics screen to identify potential biomarkers, and then done an extensive theoretical analysis of these proteins. However, they have not performed any experiments/analysis to validate their data on another dataset/experimental system. Hence, one is left a little underwhelmed.

We make similar demands of our bioinformatics papers: Bioinformatics studies that ... must both (i) answer a biological question not considered in the original publication (if there is one) or reassess the data to arrive at a different conclusion AND (ii) comprehensively and robustly validate the findings using at least two of the following: independent public data, previously unreported clinical data and/or new experimental data.

The identification of annexins as one such biomarker compounds this issue – these proteins are affected in nearly every tumour cell. Where is the validation of these markers?

Hence, despite the new data in Figure 1, there is no attempt to validate any of these findings which is a major shortcoming of this work.

The IHC data shown lacks controls and validation of the antibodies.

The formatting of the text is odd: words are split across two lines without hyphenation making comprehension difficult."

Please revise the manuscript accordingly.

·

Basic reporting

The required improvement of the figure resolution was performed.

Experimental design

There were no comments for this part.

Validity of the findings

There were no comments for this part.

Additional comments

The authors made the suggested change in Figure 2.

---

## Round 0.3 · accepted · Accept

Dear authors, i am accepting your manuscript at this stage. But i do URGE you to fix small typos, spacing issues, language confusion during the proofreading stage. Additionally, I emphasise that you must provide images with good, in fact EXCELLENT, clarity, legible and resolution, all the figures - particularly considering the proteomics and IHC studies! Your formatting does need repair!